# Direct prediction of regulatory elements from partial data without imputation

Yu Zhang[1]*, Shaun Mahony[2]*

**1** Department of Statistics, Penn State University, University Park, Pennsylvania, United States of America,
**2** Department of Biochemistry & Molecular Biology and Center for Eukaryotic Gene Regulation, Penn State University, University Park, Pennsylvania, United States of America

* yzz2@psu.edu (YZ); mahony@psu.edu (SM)

**Data Availability Statement:** IDEAS code is available from: https://github.com/seqcode/IDEAS UCSC genome browser trackhub links (hg19 and hg38) for IDEAS segmentation using 12 marks in

## Abstract

Genome segmentation approaches allow us to characterize regulatory states in a given cell type using combinatorial patterns of histone modifications and other regulatory signals. In order to analyze regulatory state differences across cell types, current genome segmentation approaches typically require that the same regulatory genomics assays have been performed in all analyzed cell types. This necessarily limits both the numbers of cell types that can be analyzed and the complexity of the resulting regulatory states, as only a small number of histone modifications have been profiled across many cell types. Data imputation approaches that aim to estimate missing regulatory signals have been applied before genome segmentation. However, this approach is computationally costly and propagates any errors in imputation to produce incorrect genome segmentation results downstream. We present an extension to the IDEAS genome segmentation platform which can perform genome segmentation on incomplete regulatory genomics dataset collections without using imputation. Instead of relying on imputed data, we use an expectation-maximization approach to estimate marginal density functions within each regulatory state. We demonstrate that our genome segmentation results compare favorably with approaches based on imputation or other strategies for handling missing data. We further show that our approach can accurately impute missing data after genome segmentation, reversing the typical order of imputation/genome segmentation pipelines. Finally, we present a new 2D genome segmentation analysis of 127 human cell types studied by the Roadmap Epigenomics Consortium. By using an expanded set of chromatin marks that have been profiled in subsets of these cell types, our new segmentation results capture a more complex picture of combinatorial regulatory patterns that appear on the human genome.

## Author summary

Histone modifications and other gene regulatory signals can be profiled across the genome in a given cell type, and each type of regulatory signal correlates with the presence of specific gene regulatory activities. Genome segmentation methods look for patterns across combinations of regulatory signals to annotate more general "regulatory states"

127 ROADMAP cell types are available from: https://github.com/seqcode/IDEAS-trackhubs.

**Funding:** This work was funded by National Institutes of Health under grant NIGMS R01GM121613 (to YZ and SM). The funders had no role in study design, data collection and analysis, decision to publish, or preparation of the manuscript.

**Competing interests:** The authors have declared that no competing interests exist.

(e.g. enhancers, promoters, repressed regions, etc.) across the genome. To see how regulatory states change across cell types, we need to run genome segmentation in a consistent way across the analyzed cell types. However, due to experimental and cost limitations, we may not have profiled the same regulatory signals in all available cell types. Current approaches deal with this missing data problem by either limiting genome segmentation analysis to the subset of regulatory signals that have been profiled in all analyzed cell types (which limits the types of regulatory states that can be detected and/or the numbers of cell types that can be analyzed), or by predicting what the missing regulatory signals would have looked like. The latter "imputation" approach is computationally costly, and is not always accurate. The current manuscript introduces a third strategy to handling missing data in the genome segmentation problem. Our approach, based on the IDEAS genome segmentation platform, removes the need for data imputation by directly accounting for missing data within the algorithm. In cell types where some regulatory signals are missing, IDEAS can still provide accurate regulatory state annotations based on a combination of the regulatory signals that have been observed in that cell type, the regulatory states annotated at the same location in other cell types (which may be based on more complete regulatory signal information), and the regulatory states in surrounding regions.

## Introduction

The combinatorial activities of regulatory elements along the genome defines cellular phenotypes during development and disease. Thanks to the proliferation of genomic assays based on massively parallel sequencing technologies, we can now comprehensively characterize genomic regulatory components by using thousands of regulatory genomic datasets generated in hundreds of cell types in human and mouse genomes. A pair of major challenges focus on identifying interpretable regulatory events across the genome and characterizing how those regulatory events vary across cell types to affect expression and phenotype. A popular solution is genome segmentation [1–3], which assigns chromatin states to genomic loci that exhibit unique combinatorial patterns of chromatin marks. The inferred chromatin states are low-dimensional de-noised representations of raw regulatory signals that produce interpretable catalogs of regulatory events in the genome. Chromatin states are valuable for studying gene regulation and disease, and hypotheses about regulatory relationships based on these states have been confirmed by functional experiments [4]. Chromatin states have also been increasingly adopted as a powerful resource for prioritizing and interpreting disease non-coding variants [5–9].

While existing genome segmentation methods have produced high quality annotations of regulatory elements in the genome, they typically require the input chromatin marks to be commonly available in all cell types of interest. When profiled chromatin marks do not match across cell types, only the common set of marks may be used for annotation. In practice, different chromatin marks are often quantified in different cell types, as they may be produced by different labs in different studies, or there may be limited budgets, different technical requirements, or the material may be unavailable. Even the largest international consortia like ENCODE [10], Roadmap Epigenomics [11], and IHEC [12] have a large proportion of missing data in their corresponding studies in most cell types. For example, while the Roadmap Epigenomics project characterized a selection of five histone modifications in all of 127 human cell types, an additional 26 histone modifications were assayed in only small subsets of the same cell types.

New methods are urgently needed to make use of all the available regulatory data sets in addition to the commonly available chromatin marks to improve the characterization of regulatory events. A current solution is to impute the missing chromatin marks before segmentation, and a few machine learning methods have been developed and show promising results [13–16]. The imputation-based approach, however, may introduce errors and biases from imputation that may have significant impact on genome segmentation results. This issue is particularly concerning when the number of missing marks is large, as is the common case in practice. Imputation also consumes a substantial amount of computing power and storage, which inevitably increases the computation burden of genome segmentation.

In this study, we introduce a new genome segmentation approach that is completely void of missing data imputation. By leveraging information across cell types, we directly segment the genomes from incomplete sets of chromatin marks. We show that not only can the new method accurately predict regulatory events from partial data, but the results also compare favorably to the imputation-based methods. Since data imputation is typically computationally much more intensive than genome segmentation, our method provides a more efficient and direct solution to the genome segmentation problem, and simultaneously will not be biased by imputation errors. Our method can still impute missing data sets as an end product; i.e. after genome segmentation. We demonstrate that this approach to data imputation can achieve comparable accuracy to that of current state-of-the-art machine learning imputation algorithms. In addition, our method can leverage existing segmentation results, such as those published on the 127 human epigenomes from the Roadmap Epigenomics Consortium, to make powerful predictions in new cell types while minimizing additional data production requirements. Importantly, our method scales linearly with respect to the number of cell types analyzed, and hence it can be realistically applied to thousands of publicly available data sets to produce comprehensive, consistent and high-quality maps of regulatory elements in all cell types.

We applied our method to the Roadmap Epigenomics data sets in 127 cell types, using 12 chromatin marks which have not been uniformly profiled across all cell types. Compared to our previous results on the data using only 5 commonly available marks, our new results contain previously unrecognized states and sub-states. Therefore, by incorporating an expanded set of chromatin marks (albeit incompletely profiled across cell types), we can potentially increase the complexity of regulatory states annotations across the genome and between cell types.

## Results

### Handling missing data in the IDEAS framework

We previously introduced IDEAS [17,18], a Bayesian nonparametric *i*ntegrative and *d*iscriminative *e*pigenome *a*nnotation *s*ystem for genome segmentation across multiple cell types simultaneously. The model has two unique features that are not considered by alternative segmentation methods. First, IDEAS identifies local cell type clusters based on similarity of local chromatin landscapes. This step identifies co-occurrence of regulatory events in local genomic intervals and borrows information from epigenetically and locally similar cell types to improve prediction. Secondly, IDEAS classifies genomic positions into categories by identifying distinct, recurring and locus-specific epigenomic profiles from all cell types. This step captures the fact that most genomic regions are epigenetically correlated across cell types due to the shared underlying DNA sequences, and the classified position categories improve the power for detecting locus-specific regulatory events. This latter feature is critically useful for the new method discussed in this study, as locus-specific modeling enables us to distinguish the

different chromatin states that may otherwise be ambiguous using the partially observed chromatin marks alone. For instance, if we only observe DNase I hypersensitivity data in a given cell type, each DNase hypersensitive location may be equally assigned to one of the several possible active regulatory states that display hypersensitivity. By locus-specific modeling, we can make a most probable guess about regulatory state membership by checking the regulatory states observed at the same site in other related cell types. An illustration of the IDEAS method is shown in Fig 1.

The IDEAS model is a fully probabilistic Bayesian mixture model, and it connects to data only through its density function in each chromatin state (a mixture component). We therefore address the missing data problem in the density functions. For density functions in the exponential family, Gaussian in our case, we can easily derive analytical marginal forms on the observed marks from the full density functions on all marks. We replace the full probability density functions by their marginal forms in the IDEAS model, and we estimate the parameters of marginal density functions from the partial data by using an expectation-maximization (EM) algorithm [19]. The EM procedure does not require data imputation at individual sites, so it can be performed very quickly. We first identify all missing data configurations in all cell types. For each configuration of data missingness, we use the EM algorithm to calculate the expected values of the sufficient statistics for the density functions (for Gaussian densities, the sums and sum of squares). We then add the estimated sufficient statistics over all configurations of missingness and estimate the model parameters by maximizing the summed sufficient statistics. This procedure guarantees a positive definite data covariance matrix, and is most efficient in theory.

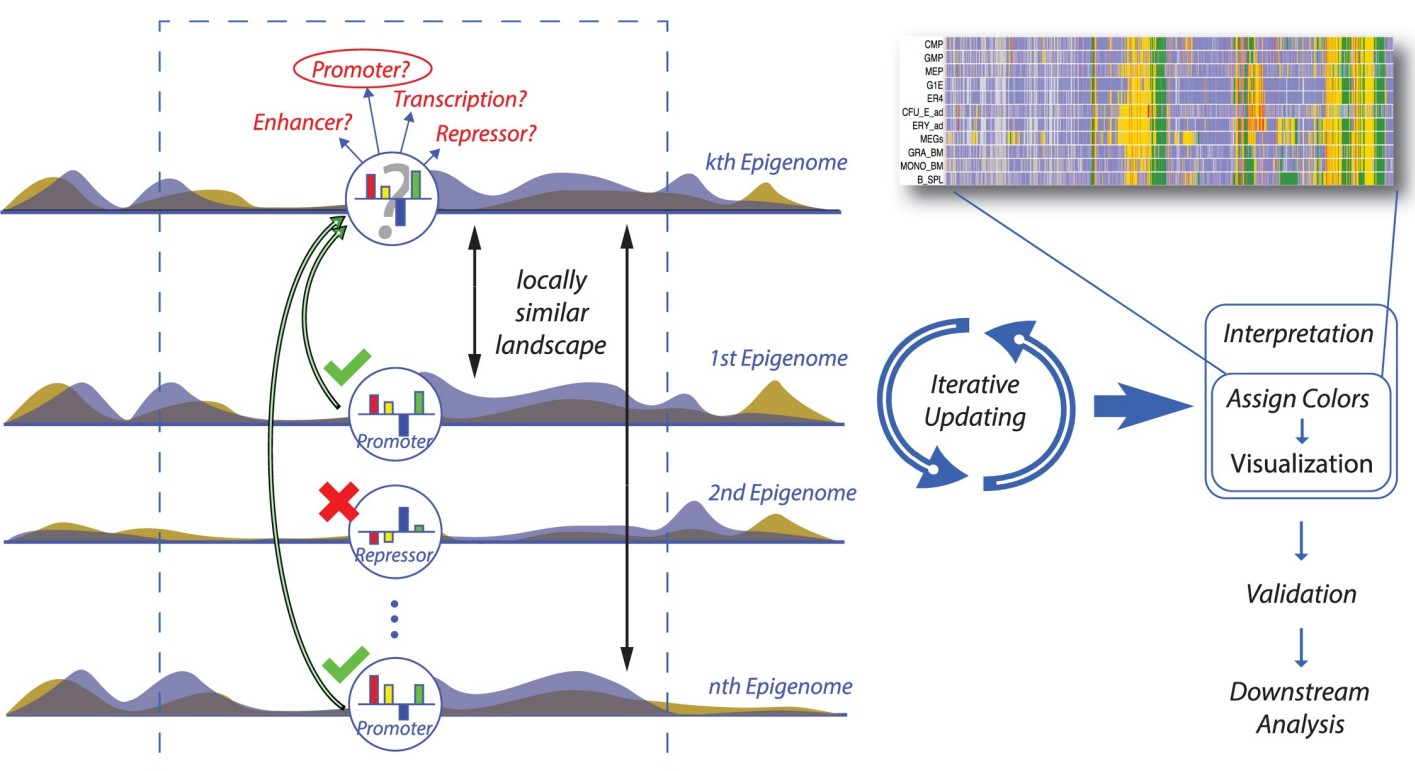

**Fig 1. Illustration of the IDEAS method.** When making an inference at a locus in a target cell type, IDEAS first identifies a set of cell types that share locally similar chromatin landscapes with the target cell type. Then IDEAS makes predictions based on the chromatin marks in the target cell type and the predictions made in the locally related cell types at the same locus. The IDEAS algorithm is a full Bayesian nonparametric probabilistic model. All model parameters, except for hyper parameters, are learned from the data, including number and parameters of chromatin states, size of local intervals, number of cell type clusters, and locus-specific profiles.

To address the data imbalance problem, where information held by cell types with just one or two marks is substantially less than the information held by cell types with many marks, we assign weights to each cell type proportional to the number of observed marks in the cell type. These weights are helpful for the model to learn more from the cell types with more data. While our solution is a standard EM algorithm for missing data problems, it enables IDEAS to directly make predictions on partial data without imputation. Unlike other segmentation tools, the unique feature of IDEAS for modeling locus-specific events, and doing so in a full probabilistic model, can gain substantial power in making predictions from partial data, and at the same time ameliorate the bias from data imbalance.

## IDEAS segmentation without imputation performs favorably compared with imputation-based approaches

The focus of this study is to evaluate the power of including partially available data in segmentation. We therefore compared our proposed approach with several other approaches to handling partial data based on a single segmentation algorithm so that differences between segmentation algorithms will not confound the comparison results. We choose to use the IDEAS method as the basis for comparison for three reasons: 1) our previous studies [17,18] have shown uniformly superior power of IDEAS over existing segmentation tools on predicting a wide variety of genomic features, such as gene expression, DNA methylation, annotated enhancers, GWAS variants, eQTLs, sequence-derived functional scores, and chromatin interaction data; 2) the proposed direct imputation method on partial data is feasible only within the IDEAS framework; and 3) the concatenation approach used in our simulation study can be treated as a proxy for the other segmentation algorithms that do not take positional information into account.

We used 17 cell types from the Roadmap Epigenomics project to evaluate our method, as these 17 cell types have 12 histone marks in common. We used the segmentation generated by IDEAS on the full data set as a benchmark for the same reasons enumerated in the previous paragraph. To create data collections with missing data, we randomly selected some cell types (1, 5 or 10 cell types) and removed a set of chromatin marks (8, 10 or 11 marks) in those cell types. The resulting partial data sets have 4%~54% missingness.

We ran several IDEAS-based approaches to handling the partial data set for the purpose of comparison: 1) direct segmentation on partial data, which pools all available unmatched marks together for joint segmentation; 2) two-step segmentation, which first segments the cell types with full marks, and then segments the remaining cell types with missing data by using the first segmentation result as a prior; this uses existing results to make predictions in new cell types; 3) segmentation via imputation by using ChromImpute [13] to impute missing data; 4) direct segmentation but with cell types concatenated side-by-side, so that no position-specific information is used; and 5) segmentation using only the commonly available marks. Approaches 1 and 2 are unique to our methods, while approaches 3, 4, and 5 are already achievable by existing methods.

We used the adjusted rand index (ARI) to measure the agreement between segmentations on partial data with segmentations on full data. ARI measures similarity between two clustering results while adjusting for different numbers of clusters and unbalanced cluster sizes, and thus is appropriate for comparing segmentation results. While we use IDEAS segmentations on full data as the benchmark, we acknowledge that all methods contain limitations and therefore IDEAS segmentation may not be perfect. However, our rationale is that we should not expect inaccuracies in IDEAS segmentation on the full data to be corrected by IDEAS-based segmentation strategies on partial data. In other words, we assume that the best that any of the partial data strategies can do is to replicate the results of IDEAS on complete data.

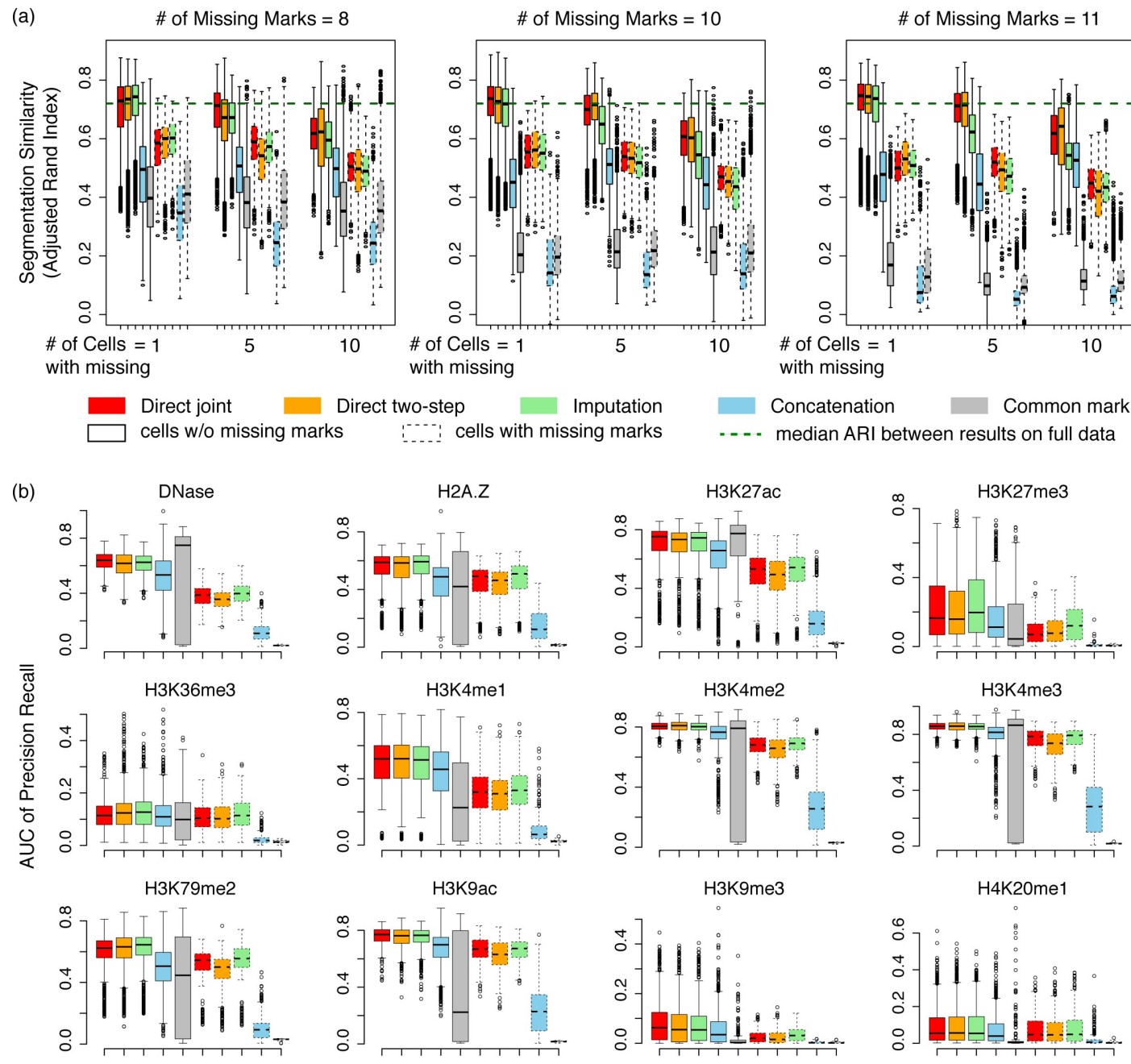

**Fig 2. Comparison of segmentation accuracy.** (a) Similarity (y-axis) of segmentation produced by different methods on partial data compared with the results obtained from the full data. Results are separated by the number of cell types with missing marks and the number of missing marks in each cell type. Results are further separated by cell types without missing marks (solid boxes) and cell types with missing marks (dashed boxes). Green dashed line indicates median similarity between independent IDEAS runs on the full data set. (b) AUC of precision-recall for predicting peaks of chromatin marks at FDR 0.05 by chromatin states. Results are separated by cell types with the mark being present (solid boxes) and cell types without the mark (dashed boxes).

As shown in Fig 2A, missing data indeed reduces the accuracy for predicting chromatin states. As expected, the more missing marks or more cell types with missing marks, the less similar segmentations are to the results derived from the full data. Comparing between the different approaches for segmenting partial data, direct segmentation on partial data without imputation, either as a joint model or a two-step approach, produced comparable or favorable

results compared with using imputed data, particularly when there are more missing marks in the data. As an empirical upper bound, the median ARI between independent runs of IDEAS on the full data is 0.73 (dashed lines in Fig 2A). As shown in the figure, when there are fewer cell types with missing data, the segmentations for cell types with full marks are not much affected by missing data in other cell types. However, when there are more cell types with missing data, even the cell types with full marks will be affected. This is because IDEAS performs joint segmentations by borrowing information across cell types. As a comparison, the concatenation approach is not highly affected by data missingness, as it does not borrow information from other cell types other than state parameters. In all cases, the concatenation approach and the approach that restricts analysis to the common marks produced less concordant results compared with the full data segmentations. This by itself does not necessarily indicate a worse performance of the common mark approach, as the input data are different. Based on the next evaluation, however, we show that restricting analysis to the common marks indeed results in a loss of information.

## Analyzing incomplete data produces richer genome segmentation results compared with restricting analysis to common marks

Many histone marks are restricted to appear in particular regulatory states. For example, H3K27me3 enrichment is restricted to those states that represent Polycomb repression, while H3K36me3 enrichment is restricted to states representing transcriptional elongation. Intuitively, accurate genome segmentation results should be somewhat predictive of the locations of chromatin marks, even if those marks were not observed. We therefore evaluated how well the chromatin states inferred by IDEAS capture the peaks of each chromatin mark. We calculated the area under curve (AUC) of precision-recall for each chromatin mark, where we ranked each chromatin state by their mean signal of the mark from high to low, and we used peaks at false discovery rate 0.05 as a reference.

As shown in Fig 2B, overall the inferred chromatin states are predictive of peaks of chromatin marks, although less so for some broader marks such as H3K36me3, H3K27me3, H3K9me3. Comparing between methods, the direct segmentation approaches (joint segmentation or two-step segmentation) performed similarly to the imputation-based method, while concatenation and common mark approaches are much less accurate. When a chromatin mark is missing in a cell type, it is expectedly harder to predict its peaks by the chromatin states. This is however even true for the imputation-based method, where ChromImpute has used sophisticated machine learning techniques to impute the missing marks. Interestingly, while the common mark approach had almost no power to recover the peaks of unobserved marks, it also performed worse for the observed marks than the joint segmentation methods. This suggests that including additional marks, even if unmatched, can improve the power of predicting peak locations for both observed and unobserved marks. Finally, among methods with unmatched data, the concatenation method performed consistently the worst, reflecting the importance of position specificity in segmentation.

## Direct segmentation on incomplete data captures cell type relationships more robustly than imputation

Another important quantity we evaluate is the cell type relationships captured by segmentation, which may be distorted by unbalanced data missingness. For each cell type, we calculated its segmentation-based distances to all other cell types by calculating one minus ARI. We then calculated the Spearman correlation of the distance vector with that derived from the full data, which evaluates how much the ranks of cell type relationships have changed. As shown in Fig

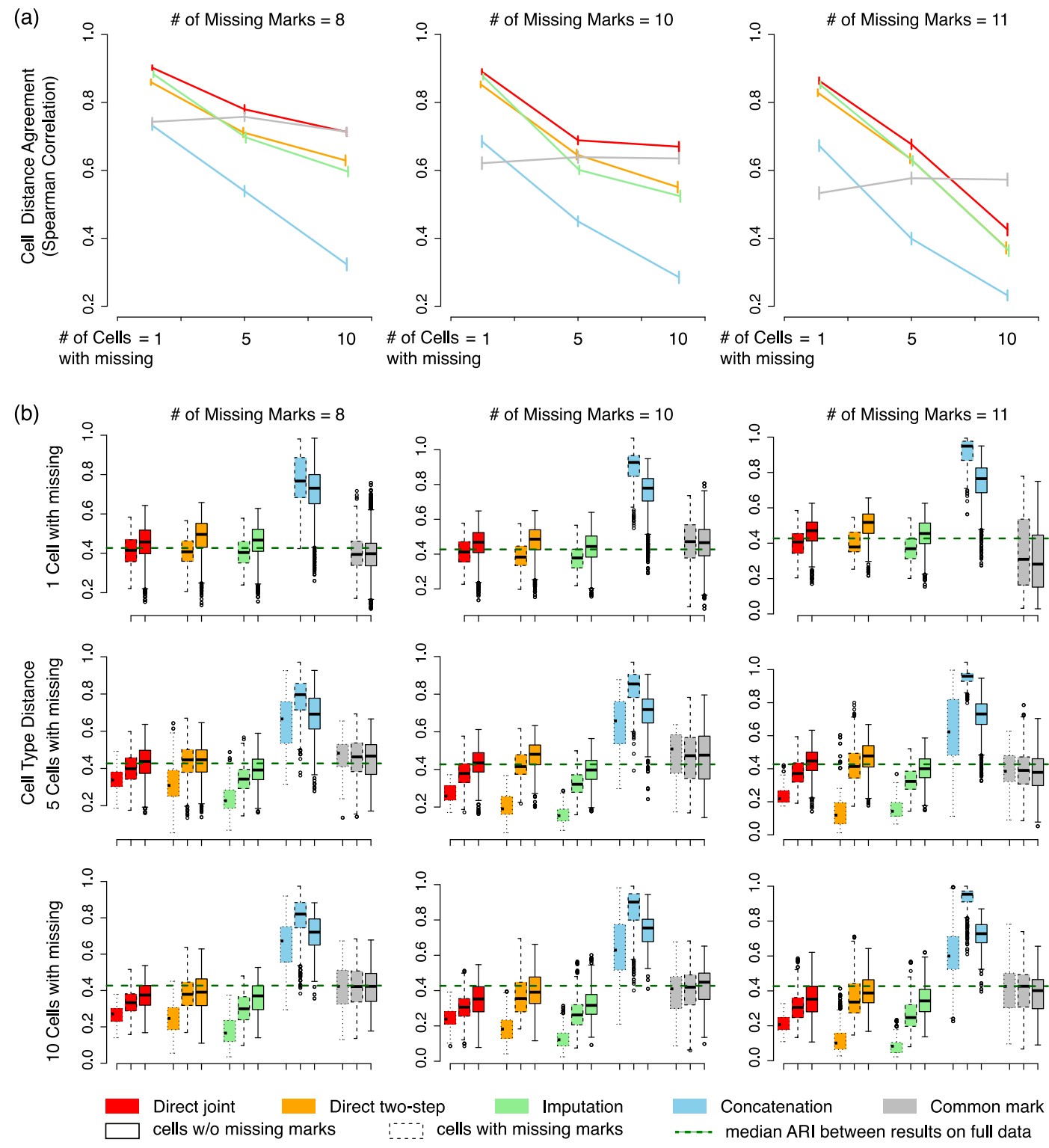

**Fig 3. Comparison of segmentation-derived cell type relationships.** (a) Concordance of cell type distances inferred from partial data by different methods compared with the distances inferred from full data segmentation. Concordance is measured by Spearman correlation between two distance vectors for each cell type. Red: joint segmentation; orange: two-step segmentation; green: imputation; blue: concatenation; grey: common mark. (b) Distribution of segmentation-based distances between

cell types with and without missing data. Dotted boxes: distances between cell types with missing data; dashed boxes: distances between cell types with missing data and cell types without missing data; solid boxes: distances between cell types without missing data. Green dashed line: median distance between cell types obtained from full data segmentations.

3A, the segmentation-based cell type relationships are affected by missing data, with more missing data leading to less agreement between the inferred cell type relationships compared with the full data cell relationships. Direct segmentation without imputation performed consistently better than the two-step approach and the imputation approach. In addition, the joint segmentation methods, with or without imputation, performed substantially better than the concatenation method. We further checked the cell type distances between cell types with missing data, between cell types with and without missing data, and between cell types without missing data, respectively (Fig 3B). Overall, the cell types with missing data are closer to each other than they are to the cell types without missing data, and the cell types without missing data have the largest distances with each other, i.e., more spread out. This bias is not surprising, as more marks can better distinguish cell types. We note however that the joint segmentation method produced the most robust cell relationships with respect to data missingness, as its cell type distances are most similar to each other among the three groups of distances. In contrast, the concatenation approach tended to create separate cell type clusters for those with or without missing data, as reflected by the greater distances between the two groups of cell types. That is, their estimated cell type relationships are dominated by data missingness rather than chromatin marks. Finally, the cell distances derived from the common mark approach was not affected by missing data, because all marks are matched between cell types.

## Quantifying the impact of missing data on segmentation accuracy

As demonstrated above, our direct and two-step approaches to segmentation on incomplete data are robust to varying degrees of data missingness. However, different types of missing data scenarios may impact segmentation accuracy more than others. For example, accurate segmentation for a cell type with missing data may require that we run segmentation alongside closely related cell types that have more complete data availability. We therefore aimed to quantify the impact of missing data on segmentation while varying one type of missing data parameter at a time.

Using the simulated partial datasets presented above (i.e. segmentation using data from 17 cell types), we examined the expected change of ARI between partial data segmentation and full data segmentation for one cell type at a time. Specifically, we regressed segmentation ARIs on 1) the number of any cell types with missing marks [$n = 1, 5, 10$]; 2) the number of cell types in the same lineage as the examined cell type with missing marks [$m = 0, 1, 2, 3$]; 3) whether or not the examined cell type has missing marks [$z = 0, 1$]; and 4) the number of missing marks in the examined cell type [$k = 8, 10, 11$]. Here, we present only the results for three blood cell types (GM12878, K562, and CD14+ Monocytes) and three lung cell types (IMR-90, A549, and NHLF), as they were the largest sets of related cell types in the 17 analyzed cell types.

As shown in Table 1, for both blood lineage and lung lineage cell types, all four types of missing data parameters had significant negative impacts on segmentation accuracy when evaluated individually. For example, looking at the coefficients of $m$ in the marginal regression setting, it may appear that segmentation accuracy depends on data availability of cell types within the same lineage. This is however unsupported by multiple regressions, in which the coefficients of $m$ are nearly insignificant. That is, the marginal impact of $m$ likely reflects the impacts of other missing parameters ($n$, $z$, or $k$). In fact, the coefficients for the other three

**Table 1. Marginal and multiple regression coefficients for blood and lung cell type segmentation accuracy when varying number of any cell type with missing data ($n$), number of cell types in same lineage with missing data ($m$), whether examined cell type has missing data ($z$), and number of missing marks in examined cell type ($k$).**

| | marginal regression coefficients | | | | multiple regression coefficients | | | |
|---|---|---|---|---|---|---|---|---|
| | $n$ | $m$ | $z$ | $k$ | $n$ | $m$ | $z$ | $k$ |
| Blood | -0.015** | -0.070** | -0.187** | -0.014** | -0.005** | -0.007* | -0.161** | -0.024** |
| Lung | -0.009** | -0.053** | -0.149** | -0.027** | -0.003** | -0.004 | -0.140** | -0.033** |

* p-value < 0.05

** p-value < 0.001

parameters are all highly significant in multiple regressions. In addition, the largest contributor to segmentation errors is $z$, whether or not the cell type of interest has missing data, while the second largest contributor (which has much less impact) is the number of missing marks in the cell type.

The above analysis suggests that cell types with fewer marks tend to be less accurately segmented than the cell types with more marks (i.e. dependence on $z$). Having more data available (either more cell types with complete data or more marks in a given cell type) generally improves segmentation performance (i.e. dependence on $n$ & $k$). This assumes that joint segmentation across cell types improves accuracy, which is supported by our previous studies in ENCODE data and Roadmap Epigenomics data [17,18]. Interestingly, it may not be as important to include a closely related cell type for joint segmentation compared with including a distantly related cell type, provided that similar amounts of data are provided overall. Of course, our study does not consider which specific epigenetic states are being inferred or which specific marks are missing, and the scale of our study is relatively small. As increasing amounts of epigenetic data sets become available, it will be of interest to perform a more comprehensive evaluation of the best strategies for genome segmentation in the context of missing data.

## IDEAS can be used to accurately impute missing marks

While we demonstrated that direct genome segmentation without imputing missing marks performed more favorably than imputation-based methods, all comparisons so far were done on the segmentation level. Segmentation-level accuracy however may not be sufficiently sensitive to reflect subtle signal variations. Fortunately, our method can also impute missing marks as an end product from the model.

We compared our imputed missing chromatin marks with those produced by ChromImpute using three measures: Pearson correlation, Spearman correlation, and mean absolute error (MAE). While Pearson correlation is fully quantitative and linear, Spearman correlation is robust to outlying values, and MAE is an overall measure of absolute difference between the predicted and the observed. As shown in Fig 4, our method and ChromImpute yielded very similar Pearson correlations with the true signals, with ChromImpute having slightly better Pearson correlations than IDEAS in a few marks such as DNase I hypersensitivity. However, using Spearman correlation we observed a different result, where IDEAS consistently outperformed ChromImpute on all marks except for DNase. This result suggests that ChromImpute may have more extreme signals in its imputed values than IDEAS does, but overall our method has better correlation with the true signals. In addition, the MAE measure showed a substantially better performance of IDEAS compared with ChromImpute in terms of the mean absolute distance to true signals. Therefore, our segmentation-first approach can be used to impute missing regulatory genomics data at least as accurately as state-of-the-art data imputation approaches.

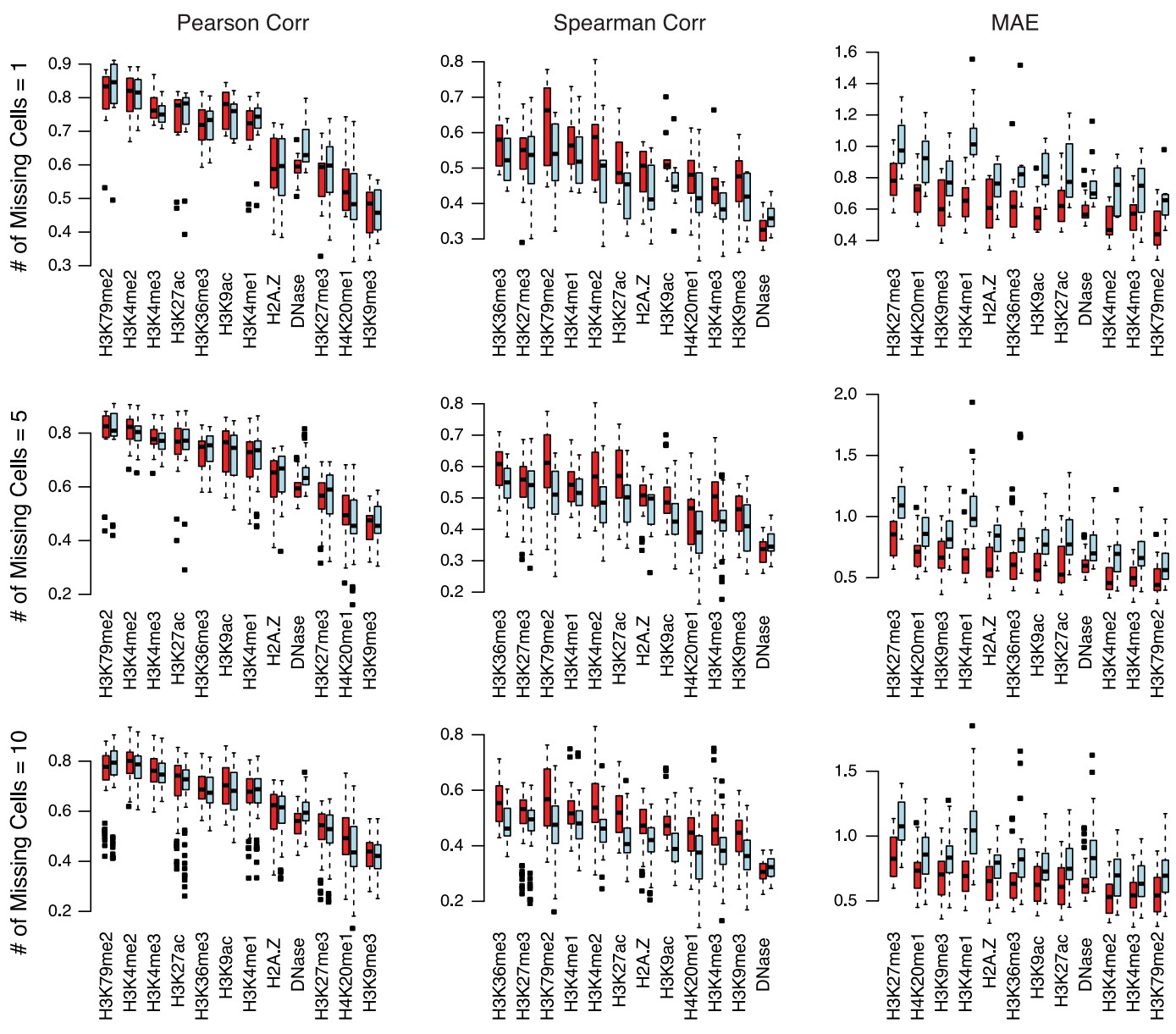

**Fig 4. Comparison of accuracy of imputed chromatin marks between IDEAS and ChromImpute.** Three accuracy measures are calculated: Pearson correlation, Spearman correlation, and mean absolute error (MAE) between IDEAS (red boxes) and true signals, and between ChromImpute (blue boxes) and true signals. The results shown are calculated from 8 missing marks per missing cell. Results for 10 and 11 missing marks are similar and shown in S1 Fig and S2 Fig, respectively.

## Segmentation of 127 Roadmap Epigenomics cell types

We applied our method to re-segment the 127 epigenomes from the Roadmap Epigenomics project. In addition to using the five commonly available marks (H3K4me3, H3K4me1, H3K36me3, H3K27me3, H3K9me3), which we have used to generate a previous map in these epigenomes [18], we added 7 new marks (H3K9ac, H3K4me2, H2A.Z, DNase, H3K27ac, H3K79me2, H4K20me1) that are only available in a small portion of the 127 cell types (Fig 5A). We ran IDEAS on the 12 chromatin marks to produce a 42-state model, where the

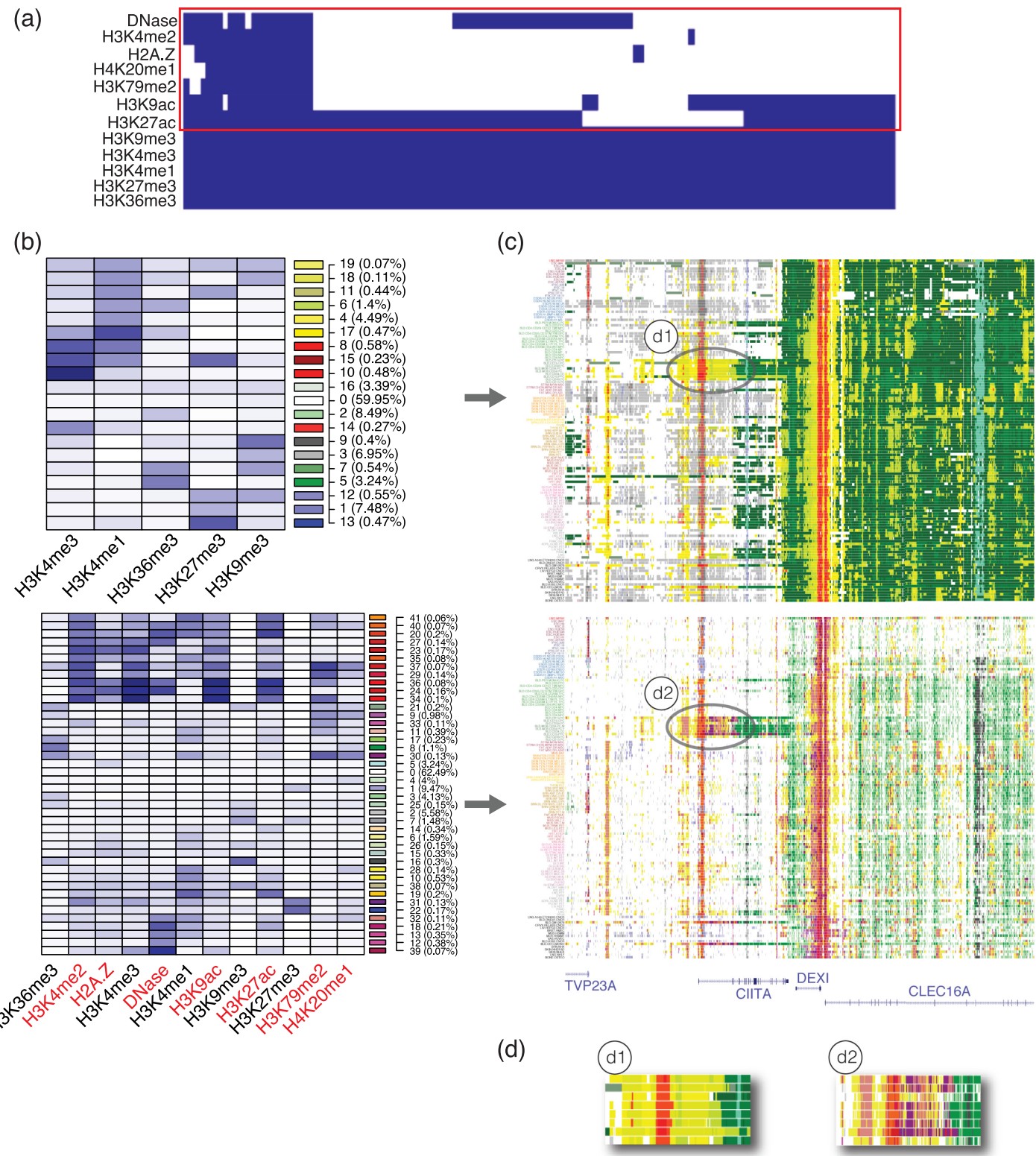

**Fig 5. Segmentation of 127 Roadmap Epigenomics cell types using 12 marks.** (a) Presence (blue) and absence (white) of the 12 marks used in the new segmentation. Red box indicates the 7 new marks. (b) Mean signals of chromatin states in the previously published 5-mark map (top) compared with the mean signals of chromatin states inferred using 12 marks (bottom). State colors used in genome browser are shown on the side in each row, along with state ID and proportion of the state in the genome. (c) Example browser view of the previous map compared to the new map at the CIITA gene (hg19 chr16:10,900,000–11,150,000). (d) Zoomed in views at CIITA promoter.

algorithm using a 50% reproducibility threshold automatically determined the number of states. Compared with the previous map generated by IDEAS on 5 marks, the new map identified sub-states of the existing states as well as new states specific to the 7 new marks (Fig 5B). An example of the UCSC genome browser view of the segmentation results at the *CIITA* gene is further shown in Fig 5C & 5D, where subtle details of chromatin states are revealed in the new map, especially at the promoter region of *CIITA*. UCSC genome browser trackhub links of the new map generated by IDEAS using 12 marks are available for hg19 and hg38 at: https://github.com/seqcode/IDEAS-trackhubs.

## Expanded segmentation on partial data provides improved characterization of regulatory events

We next aimed to assess whether our new 127 cell segmentations using expanded partial data (12 marks) represent regulatory events more accurately than the original segmentation using complete data (5 marks). We therefore asked how well the chromatin state annotations can predict gene expression levels and active enhancer locations (where the latter is defined as the locations of enhancer RNA enrichment).

First, we used the chromatin states to predict $\log_2$ transformed RNA-seq RPKM values from 56 Roadmap Epigenomics cell types. We performed two types of regression analysis: 1) predicting expression across genes in each cell type; and 2) predicting expression across cell types for each gene. In both cases, we first converted the multi-dimensional categorical chromatin state profile within each region of interest into one-dimensional numerical predictors that integrate the associations between each state's presence at transcription start sites (TSSs) and expression levels (see Methods). We included two types of regions in our study: 1) regions within 2kbp of TSSs; and 2) regions within 500kbp but beyond 2kbp of TSSs (distal regions). We obtained one numerical predictor for each type of region, and thus our predictor space has at most two-dimensions.

In predicting expression across genes in each cell type, we found that the chromatin states at TSS regions were highly predictive of gene expression, with mean $R^2$ = 52.1% for partial 12-mark segmentation ($R^2$ was calculated in each cell type separately and averaged), and 52.6% for complete 5-mark segmentation). This is not surprising, as the variance of expression across genes is mainly driven by the on/off statuses of genes, which are readily implied by the chromatin states at TSSs. In contrast, the predictive power by distal regions alone is much lower (mean $R^2$ = 8.9% for 12-mark segmentation and 9.1% for 5-mark segmentation). In fact, when using both TSS and distal regions to predict RNA levels across genes, the mean $R^2$ did not increase much (mean $R^2$ = 52.2% for 12-mark segmentation and 52.7% for 5-mark segmentation). Comparing the fitting of two segmentations on RNA data, the two did not yield statistically significant difference (Wilcoxon sign test of the fitting $R^2$s had p-value 0.22).

In contrast, we observed a more profound impact of the distal regions on predicting gene expression across cell types, i.e., for predicting differential expression per gene. As shown in Fig 6A–6D, the partial data 12-mark segmentation indeed had improved power for predicting differential gene expressions than the complete data 5-mark segmentation, significantly so for genes with higher mean expression levels (mean $\log_2$ RPKM > -1). In general, the new segmentation improved the power for predicting differential expression by 10%~20% relative to that of the old segmentation.

We next used chromatin states from the two segmentations to predict enhancer RNA (eRNA) data from the Fantom5 project [20,21]. As described in the methods, we used chromatin states in each cell type to predict the mean enhancer signals (either eRNA read counts or peak presence) across Fantom5 libraries. As shown in Fig 6E and 6F, the partial 12-mark

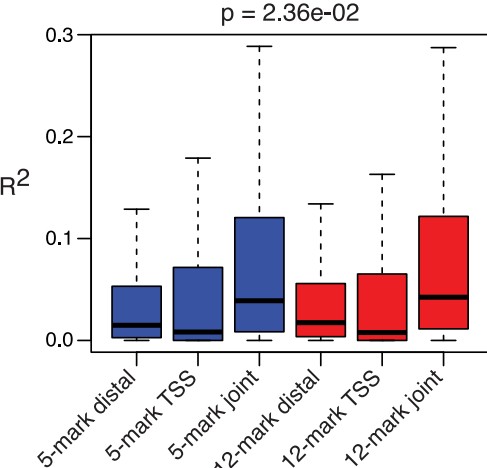

**(a) Low mean, low std. dev. genes**
(mean < -1, sd < 2)
p = 2.36e-02

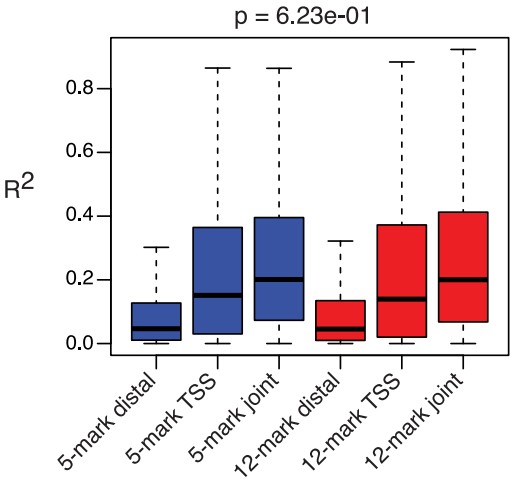

**(b) Low mean, high std. dev. genes**
(mean < -1, sd > 2)
p = 6.23e-01

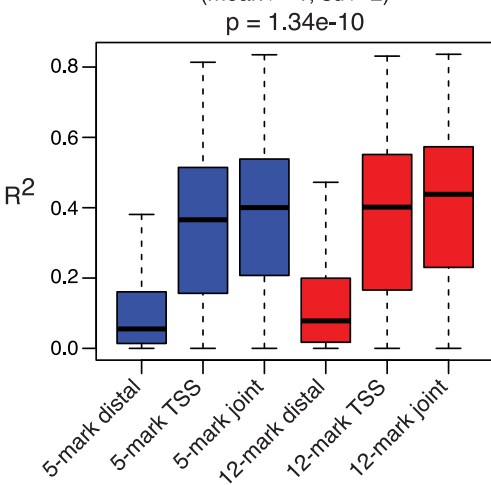

**(c) High mean, high std. dev. genes**
(mean > -1, sd > 2)
p = 1.34e-10

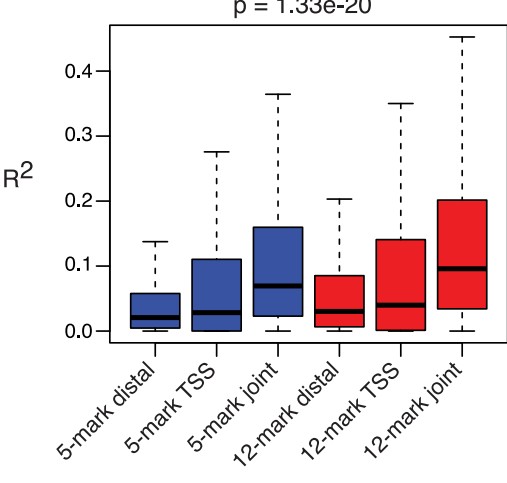

**(d) High mean, low std. dev. genes**
(mean > -1, sd < 2)
p = 1.33e-20

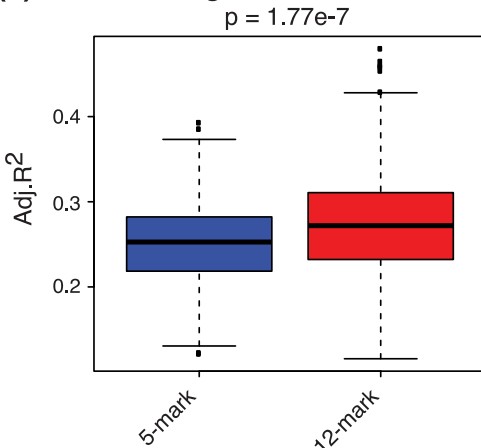

**(e) Predicting Fantom5 eRNA TPM**
p = 1.77e-7

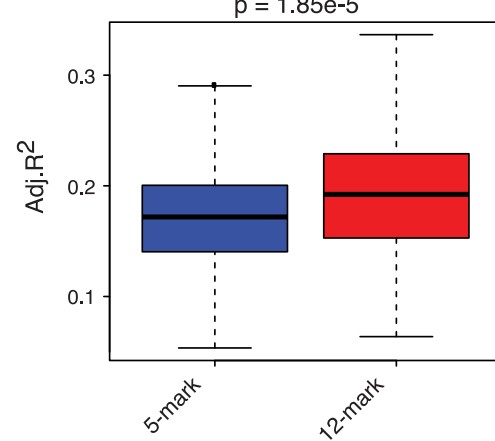

**(f) Predicting Fantom5 eRNA peaks**
p = 1.85e-5

**Fig 6. Segmentation of 127 Roadmap Epigenomics cell types using 12 marks outperforms 5-mark segmentation in explaining regulatory elements. (a-d)** 12-mark segmentation displays a greater ability to predict gene expression across cell types per gene in four separate categories of genes. **(e-f)** 12-mark segmentation outperforms 5-mark segmentation in predicting Fantom5 eRNA TPM expression levels (e), and eRNA peaks (f).

segmentation had significantly superior power for predicting enhancers compared with the complete 5-mark segmentation. Note that we evaluated the power by adjusted $R^2$, which accounts for the different numbers of epigenetic states in the two segmentations. Combined with the RNA-seq analysis, we thus conclude that the new segmentation using more (but partial) data can more accurately capture gene regulatory activities in the genome than the previous segmentation using the 5 common marks. In particular, the greater ability to predict eRNAs and differential expression when integrating distal elements suggests that the new 12-mark segmentation captures a more nuanced definition of enhancer elements.

Since the number of states are different between the two segmentations, we performed an additional experiment to check if the improved performance of the 12-mark segmentation is merely a result of its increased number of states. We ran IDEAS on the 5 common marks, but constrained the model to infer 42 states, thereby matching the number of states in the partial 12-mark segmentation. We used the resulting segmentation to predict RNA levels and Fantom5 eRNAs as before. Consistently in all comparisons (S1 Table & S2 Table), we observed inferior predictive power compared with both the original 20-state (5 common mark) model and the new 42-state (partial 12-mark) model. These results suggest that the improved predictive power of the new segmentation is attributable to the additional marks used.

## Discussion

Genome segmentation approaches have proven to be invaluable for characterizing regulatory activities across the genome. By estimating the identities and locations of regulatory states based on combinatorial patterns of chromatin marks, we can annotate genomic loci as displaying the chromatin signatures of potential enhancers, promoters, transcribed regions, repressed regions, etc. The complexity of the states, and therefore the richness of the genomic annotations, depend on the number and types of chromatin marks analyzed. For example, performing genome segmentation on data that includes the marks H3K4me1 and H3K4me3 allows us to annotate potential enhancers, but including H3K27ac alongside those marks enables us to subcategorize enhancers into active and poised sub-types [22,23].

To study how regulatory activities change across cell types and conditions, we require methods that perform genome segmentation consistently across multiple cell types. However, IDEAS and other proposed approaches for multi-cell genome segmentation have relied on the assumption that data for the same chromatin marks are available in all analyzed cell types. This inevitably limits the application of multi-cell genome segmentation. There are many cell types and many chromatin marks, but only a tiny subset of all possible experiments have been performed. While consortium efforts continue to characterize epigenomic profiles, it is unrealistic to expect that the full matrix of possible experiments will ever be completed. Therefore, newer genome segmentation methods should take advantage of all observed data to provide richer regulatory state annotations, even if some chromatin marks have not been characterized in all analyzed cell types.

Our extensions to the IDEAS platform represent the first approach for handling missing data in genome segmentation without requiring data imputation. As our results demonstrate, our direct approach performs at least as well as a comparable imputation-based strategy in controlled simulation studies with varying degrees of data missingness. Interestingly, our approach outperforms imputation in annotating cell types with no missing data in scenarios

where there is a high degree of missing data overall. This suggests that imputation may introduce unexpected forms of bias into the overall genome segmentation results. Therefore, performing inference with missing data handled by modeling appropriate marginal density functions may be a safer analysis strategy than treating imputed data the same as observed data. We also note that both our direct approach and data imputation greatly outperform simpler strategies such as restricting genome segmentation analysis to marks profiled in all cell types. This demonstrates the advantages of accounting for all observed data as opposed to focusing on more limited complete data matrices.

We compared two IDEAS-based approaches to handling missing data without imputation in this study. The first applies segmentation directly to the entire incomplete data collection, while the second performs a step-wise analysis that applies segmentation to the subset of cell types that contain all profiled marks and then uses those results as a prior for analysis of the incompletely profiled cell types. Both approaches perform similarly in most tests. Thus, our first approach suggests that direct analysis of incomplete data collections is feasible and accurate. However, our second approach also suggests an efficient strategy by which genome segmentation of well-characterized reference epigenomes may be leveraged to provide insight into the regulatory landscape of new cell types, even if only an incomplete set of chromatin marks has been profiled in those cells.

While our main goal in this study is to demonstrate an approach for genome segmentation without requiring data imputation, we also demonstrated that imputation of missing data can be performed after genome segmentation as a by-product of our approach. Interestingly, our post-segmentation approach to imputation outperforms ChromImpute in terms of Spearman correlation and mean absolute error compared with held-out observed data. However, we also note that our imputation approach is currently quite simple, as it merely applies the expected mean signal from the annotated chromatin state as the predicted value for a given missing signal. If the primary interest is to impute missing chromatin signals, we suggest that IDEAS segmentation results could be combined with analysis of local variation in observed signals in order to predict local signal levels. We also note that we have not compared our simple imputation approach to more recent imputation methods [14,15], which may offer greater performance than ChromImpute.

Finally, our extension to IDEAS is fast and scalable to large numbers of cell types and chromatin marks, as demonstrated by our application to an incomplete matrix of 12 chromatin mark experiments performed across 127 human cell types. This analysis also comes without the computational time and storage overheads associated with imputation-first strategies to genome segmentation. IDEAS can therefore provide efficient and consistent genome segmentation across large and incomplete collections of epigenomic data.

## Materials and methods

### Roadmap Epigenomics datasets

As described in our previous study [18], we downloaded the negative $\log_{10}$ of the Poisson P-value tracks for chromatin marks assayed in 127 cell types from the Roadmap Epigenome Consortium http://egg2.wustl.edu/roadmap/data/byFileType/signal/consolidated/macs2signal/pval/. The 127 cell types were chosen based on all of them sharing a core set of five chromatin marks (H3K4me3, H3K4me1, H3K36me3, H3K27me3 and H3K9me3). We processed the signal tracks of each mark by taking the mean per 200bp window across the genome in hg19. We removed regions associated with repeats and blacklisted regions as given in (http://hgdownload.cse.ucsc.edu/goldenPath/hg19/encodeDCC/wgEncodeMapability/wgEncodeDukeMapabilityRegionsExcludable.bed.gz) and (http://hgdownload.cse.ucsc.edu/goldenPath/hg19/encodeDCC/wgEncodeMapability/wgEncodeDacMapabilityConsensusExcludable.bed.gz).

We took the $\log_2(x + 0.1)$ transformation of the data as input to IDEAS. Here, $x$ denotes the negative $\log_{10}$ P-values that were provided by the Roadmap Epigenomics project. We additionally applied $\log_2$ transformation of values plus the constant 0.1 to reduce data skewness.

## Direct segmentation with missing data

The IDEAS model is a Gaussian mixture component model, with cell type and locus structures built on the distribution of mixture components. The chromatin states are the mixture components, and the chromatin marks are modeled by an emission probability function, specifically a multivariate Gaussian distribution, in each chromatin state. As in any mixture model, we only need to account for the missing data in the emission probability functions. Distinct from simple mixture models, the structures on the mixture components will help IDEAS to identify the correct components (chromatin state) when data are missing.

Let $(\mathbf{X_{obs}}, \mathbf{X_{mis}})$ denote an $n$ by $p$ matrix of data of the complete set of chromatin marks within a state. Here $n$ denotes the number of instances of the state throughout the genome in all cell types, and $p$ denotes the total number of chromatin marks. The signals of the observed marks are in $\mathbf{X_{obs}}$, and the signals of the missing marks are in $\mathbf{X_{mis}}$, which is unobserved. To model missing data in the emission probability function of the state, we need two sufficient statistics: mean and covariance, which are easily derived from the matrix

$$
\mathbf{V} = \mathbf{1}^{\mathrm{T}}\mathbf{1}, \quad \mathbf{X_{obs}}^{\mathrm{T}}\mathbf{1}, \quad \mathbf{X_{mis}}^{\mathrm{T}}\mathbf{1}
$$
$$
\mathbf{1}^{\mathrm{T}}\mathbf{X_{obs}}, \quad \mathbf{X_{obs}}^{\mathrm{T}}\mathbf{X_{obs}}, \quad \mathbf{X_{obs}}^{\mathrm{T}}\mathbf{X_{mis}}
$$
$$
\mathbf{1}^{\mathrm{T}}\mathbf{X_{mis}}, \quad \mathbf{X_{mis}}^{\mathrm{T}}\mathbf{X_{obs}}, \quad \mathbf{X_{mis}}^{\mathrm{T}}\mathbf{X_{mis}}
$$

Thus, our task is to estimate $\mathbf{V}$ from the observed data. Note that to estimate $\mathbf{V}$ we do not need to impute individual values in $\mathbf{X_{mis}}$, but only to impute the sufficient statistics (i.e., sums and sums of squares), which can be done by an Expectation-Maximization (EM) algorithm.

Here, we abuse the notation a little bit, because different cell types may have different sets of missing marks. Therefore, the notation $\mathbf{X_{obs}}$, $\mathbf{X_{mis}}$ is cell type-specific, which renders the notation in $\mathbf{V}$ meaningless. However, we use the notation $\mathbf{X_{obs}}$, $\mathbf{X_{mis}}$ to distinguish the observed and the missing marks, but keep in mind that such ordering of marks is only meaningful within a cell type. Within each cell type, we arrange marks according to which marks are observed and which are missing. After estimating the sufficient statistics for the cell type, we rearrange the matrix to match the order of marks used in $\mathbf{V}$.

## EM algorithm

We first estimate the regression coefficients in $\mathbf{X_{mis}} \sim \boldsymbol{\alpha} + \boldsymbol{\beta}\mathbf{X_{obs}}$, where the closed-form solution of $\boldsymbol{\alpha}$ and $\boldsymbol{\beta}$ can be expressed as functions of sufficient statistics. Since we do not know $\mathbf{X_{mis}}$, we start with an initial guess $\mathbf{V_0}$, which is an identity matrix in our case, so that we can obtain $\boldsymbol{\alpha_0}$ and $\boldsymbol{\beta_0}$ from $\mathbf{V_0}$ without knowing $\mathbf{X_{mis}}$. Of course, our initial estimates $\boldsymbol{\alpha_0}$ and $\boldsymbol{\beta_0}$ are inaccurate at best. We will repeat the EM steps until the algorithm converges to the maximum likelihood estimates (MLE).

For each cell type $k$, let $\mathbf{X^k_{obs}}$, $\mathbf{X^k_{mis}}$ denote the observed and missing data matrix in cell type $k$ in the specific chromatin state. We calculate

$$
\mathbf{X^k_{mis}}^{\mathrm{T}}\mathbf{1} = (\boldsymbol{\alpha_0} + \boldsymbol{\beta_0}\mathbf{X^k_{obs}})^{\mathrm{T}}\mathbf{1}
$$
$$
\mathbf{X^k_{mis}}^{\mathrm{T}}\mathbf{X^k_{obs}} = (\boldsymbol{\alpha_0} + \boldsymbol{\beta_0}\mathbf{X^k_{obs}})^{\mathrm{T}}\mathbf{X^k_{obs}}
$$
$$
\mathbf{X^k_{mis}}^{\mathrm{T}}\mathbf{X^k_{mis}} = (\boldsymbol{\alpha_0} + \boldsymbol{\beta_0}\mathbf{X^k_{obs}})^{\mathrm{T}}(\boldsymbol{\alpha_0} + \boldsymbol{\beta_0}\mathbf{X^k_{obs}})
$$

which avoids imputing missing data at the individual level. This gives an estimate of $\mathbf{V}$ from cell type $k$, denoted as $\mathbf{V^k_1}$, where subscript 1 is the current iteration number (0) plus 1.

We repeat the above step for all cell types $k = 1, \ldots, K$, then an updated estimate of $\mathbf{V}$ is given by

$$\mathbf{V_1} = \sum_k \mathbf{V^k_1}$$

which completes one EM iteration for one state.

We repeat the EM iterations until the estimated sufficient statistics matrix series $\mathbf{V_0}, \mathbf{V_1}, \mathbf{V_2}, \ldots$, converges, which yields the maximum likelihood estimate of mean and covariance parameters for the Gaussian distribution in the state. We repeat this algorithm to obtain MLEs for all chromatin states. Finally, given the multivariate Gaussian parameters, marginal emission probability functions for the observed marks are easily derived, which completes the IDEAS model with missing data.

## Imputing individual marks

While IDEAS does not need to impute individual signals of the missing marks, the signals can be imputed as the mean signal of the mark from the emission distribution of the corresponding chromatin state.

## Simulation procedure

We used 12 chromatin marks in 17 cell types from the Roadmap Epigenomics project to carry out simulation studies. The 12 marks are H3K4me3, H3K4me1, H3K36me3, H3K27me3, H3K9me3, H3K9ac, H3K4me2, H2A.Z, DNase, H3K27ac, H3K79me2, H4K20me1, and the 17 cell types are H1-hESC (E003), H9-hESC (E008), IMR-90 (E017), A549 (E114), GM12878 (E116), HeLa-S3 (E117), HepG2 (E118), HMEC mammary epithelial (E119), HSMM skeletal muscle myoblasts (E120), HSMM skeletal muscle myotubes (E121), HUVEC (E122), K562 (E123), Monocytes-CD14+ (E124), NH-A astrocytes (E125), NHDF-Ad (E126), NHEK (E127), NHLF (E128). We used these data because the 12 marks are commonly available in the 17 cell types.

To generate a simulated data set with missing marks, we first randomly sampled a given number of marks and a given number of cell types. We removed the sampled marks from each of the sampled cell types. The remaining data are treated as observed, and the removed data are used for validation.

## Running IDEAS

We ran IDEAS on the simulated data set in five ways to represent different segmentation strategies: 1) Directly segment the data set with missing marks; 2) First perform segmentation in those cell types with the complete set of marks, and then segment the remaining cell types with missing marks by using the first segmentation result as priors. In particular, the sufficient statistics (means and covariance matrices) of the chromatin states obtained in the first step are input to the second step as Gaussian priors for the corresponding chromatin states; 3) Run ChromImpute [13] in its default setting to impute the missing marks first, and then run IDEAS as if the missing marks are observed; 4) Concatenate the cell types side-by-side as if the data from different cell types are generated from a single pseudo genome, so that no position-specific information is used. We then run IDEAS on the concatenated genome with missing marks; And 5) perform segmentation using only the commonly available marks.

In all cases, IDEAS was run using 200bp windows across the genome. IDEAS has a built-in function to evaluate reproducibility of chromatin states from random regions in the genome. The number of chromatin states is thus automatically determined by two factors: a Bayesian penalty on the number of chromatin states, and a 50% reproducibility threshold at which the chromatin states must be reproducible with 50% probability between two independent runs.

We further assigned a weight to each cell type proportional to the number of observed marks in the cell type. In particular, $w_i$ for cell type $i$ is 1 if all marks are observed, and $w_i = 1/12$ if only one out of 12 marks is available. Accordingly, within the IDEAS model, each chromatin state inferred in the cell type $i$ will be treated as a fractional ($w_i$) observation. The weight only applies to the inference of chromatin states, as missing data is at the cell type level, and it does not apply to the Gaussian parameter estimation within states.

## Calculation of area under precision recall curves

We calculated AUC of precision recall curves by comparing the inferred chromatin states with peak calling results. Specifically, we called peaks at 200bp windows using a z-test assuming Gaussian distribution of the log transformed data, where the mean and the standard deviation of the null distribution (i.e., no peaks) were estimated using the whole genome data within each cell type. We then calculated the p-value of the z-score in each 200bp window, and finally called peaks at genome-wide FDR 0.05. Given the peaks, we calculated precision recall values by ranking the chromatin states in a decreasing order of the mean signal of the corresponding chromatin mark. Pretending that the first $k$ chromatin states were used to predict peaks, we overlapped the instances of these $k$ states with the peaks and calculated precision and recall values. We repeated this for $k = 1, \ldots, K$, where $K$ denotes the total number of chromatin states, which created a precision-recall curve, from which we finally calculated the area under the curve.

## Prediction of RNA-seq levels and enhancer elements

RNA-seq RPKM (read counts per kilobase of exon per million reads) values are sourced from the Roadmap Epigenomics Project website. We converted the multi-dimensional categorical chromatin state profile within each region of interest into one-dimensional numerical predictors by regressing RNA-seq data on the percentage of occurrence of each chromatin state within 2kbp regions centered on the transcription start sites of genes, across all genes and all cell types. The regression coefficient for each epigenetic state is then multiplied to the percentage of the state appearing in each region of interest to obtain a numeric value. For example, if the coefficients for state A and state B are 3 and -2, respectively, and within a region of interest there are 20% state A and 80% state B, then the numeric value assigned to the region is 3*0.2 + (-2)*0.8 = -1. In this way, all epigenetic states are converted to a one-dimensional numerical vector as a quantitative predictor of gene expression, regardless of the original dimension of the epigenetic states.

When predicting expression across cell types per gene, we split the genes into four categories: 1) low mean & low standard deviation (mean $\log_2$ RPKM $< -1$, sd $< 2$), representing genes that are off in most examined cell types; 2) low mean & high standard deviation (mean $\log_2$ RPKM $< -1$, sd $> 2$), representing genes that are lowly expressed in most examined cell types, but show some differential expression; 3) high mean & high standard deviation (mean $\log_2$ RPKM $> -1$, sd $> 2$), representing genes that show high expression in a subset of cell types; and 4) high mean & low standard deviation (mean $\log_2$ RPKM $> -1$, sd $< 2$), representing genes that are on in most examined cell types. The significance values in Fig 6 are calculated by Wilcoxon sign test on subsampled genes (n = 3,872) so that each category of genes have the same sample size.

We obtained both eRNA TPM (tag counts per million) data and the called enhancer peaks from the Fantom5 website [20,21]. There are data from 808 libraries, which do not have one-to-one map to Roadmap Epigenomics cell types. We therefore calculated mean values across the 808 libraries as our reference enhancer signals. In particular, for the eRNA TPM data, we took mean of the $\log_2$ TPM values due to data skewness. For the enhancer peak data, we took arithmetic mean of the binary presence/absence values across libraries and then applied the logit transformation. We then used chromatin states in each cell type to predict the mean enhancer signals.

## Supporting information

**S1 Table. Mean $R^2$ of gene expression prediction using both TSS and distal regions.**
(DOCX)

**S2 Table. Mean $R^2$ of eRNA prediction.**
(DOCX)

**S1 Fig. Comparison of accuracy of imputed chromatin marks between IDEAS and ChromImpute.** Three accuracy measures are calculated: Pearson correlation, Spearman correlation, and mean absolute error (MAE) between IDEAS (red boxes) and true signals, and between ChromImpute (blue boxes) and true signals. The results shown are calculated from 10 missing marks per missing cell.
(EPS)

**S2 Fig. Comparison of accuracy of imputed chromatin marks between IDEAS and ChromImpute.** Three accuracy measures are calculated: Pearson correlation, Spearman correlation, and mean absolute error (MAE) between IDEAS (red boxes) and true signals, and between ChromImpute (blue boxes) and true signals. The results shown are calculated from 11 missing marks per missing cell.
(EPS)

## Author Contributions

**Conceptualization:** Yu Zhang.

**Formal analysis:** Yu Zhang.

**Funding acquisition:** Yu Zhang, Shaun Mahony.

**Investigation:** Yu Zhang.

**Methodology:** Yu Zhang.

**Project administration:** Shaun Mahony.

**Software:** Yu Zhang.

**Supervision:** Shaun Mahony.

**Writing – original draft:** Yu Zhang, Shaun Mahony.

**Writing – review & editing:** Yu Zhang, Shaun Mahony.

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
