## [Decision Letter · Decision Letter 0]

21 Jun 2019

Dear Dr Mahony,

Thank you very much for submitting your revised manuscript 'Direct prediction of regulatory elements from partial data without imputation' for review by PLOS Computational Biology. Your manuscript has been fully evaluated by the three independent peer reviewers who had reviewed the first version of this manuscript when submitted to GLBIO.  While two of the reviewers are satisfied with the current version of the work, reviewer 2's comments still need some additional work. While your manuscript cannot be accepted in its present form, we are willing to consider a revised version in which the issues raised by reviewer 2 have been adequately addressed. 

Sincerely,

Sushmita Roy, Ph.D.

Associate Editor

PLOS Computational Biology

William Noble

Deputy Editor

PLOS Computational Biology

[LINK]

Reviewer's Responses to Questions

**Comments to the Authors:**

Reviewer #1: My original review was positive and I do not have further comments.

Reviewer #2: The authors have address my concerns.

Reviewer #3: The addition of two different sections and some text to clarify limitations (I appreciate the truthful statement about the imputation part in the Discussion) have strengthened the manuscript. There are two points that are not fully addressed either because of time limitation or misunderstanding of my suggestions, which would be great to do so before publication.

- For my comment on using only previous iteration of IDEAS results as gold standard, for the completeness of this manuscript, that would be important to add a summary of the comparative results of IDEAS and other methods from the previous work. That would justify, to some extent, the use of IDEAS-only evaluation together with the added text.

- The comment on "why more states are better" was mainly referring to a controlled experiment where the number of states are set to an equal number for both the complete 5-mark segmentation and the incomplete 12-mark. The evaluations that are done would still be applicable but differences could then directly be attributed to using more marks including those that are complete rather than, say, more granular state assignments due to simply having more marks (i.e., 5 marks are not rich enough to have 42 meaningfully distinct states but 12 with imputation are).

**Have all data underlying the figures and results presented in the manuscript been provided?**

Reviewer #1: Yes

Reviewer #2: Yes

Reviewer #3: No: Supplementary tables need to be provided with segmentation information from each dataset for reproduction of at least each main figure.

PLOS authors have the option to publish the peer review history of their article (what does this mean?). If published, this will include your full peer review and any attached files.

Reviewer #1: No

Reviewer #2: No

Reviewer #3: Yes: Ferhat Ay

---

## [Decision Letter · Decision Letter 1]

12 Sep 2019

Dear Dr Mahony,

We are pleased to inform you that your manuscript 'Direct prediction of regulatory elements from partial data without imputation' has been provisionally accepted for publication in PLOS Computational Biology.

In the meantime, please log into Editorial Manager at https://www.editorialmanager.com/pcompbiol/, click the "Update My Information" link at the top of the page, and update your user information to ensure an efficient production and billing process.

One of the goals of PLOS is to make science accessible to educators and the public. PLOS staff issue occasional press releases and make early versions of PLOS Computational Biology articles available to science writers and journalists. PLOS staff also collaborate with Communication and Public Information Offices and would be happy to work with the relevant people at your institution or funding agency. If your institution or funding agency is interested in promoting your findings, please ask them to coordinate their releases with PLOS (contact ploscompbiol@plos.org).

Thank you again for supporting Open Access publishing. We look forward to publishing your paper in PLOS Computational Biology.

Sincerely,

Sushmita Roy, Ph.D.

Associate Editor

PLOS Computational Biology

William Noble

Deputy Editor

PLOS Computational Biology

Reviewer's Responses to Questions

**Comments to the Authors:**

Reviewer #3: All comments addressed.

**Have all data underlying the figures and results presented in the manuscript been provided?**

Reviewer #3: Yes

PLOS authors have the option to publish the peer review history of their article (what does this mean?). If published, this will include your full peer review and any attached files.

Reviewer #3: Yes: FERHAT AY

---

## [Editor Report · Acceptance letter]

4 Oct 2019

PCOMPBIOL-D-19-00827R1 

Direct prediction of regulatory elements from partial data without imputation

Dear Dr Mahony,

I am pleased to inform you that your manuscript has been formally accepted for publication in PLOS Computational Biology. Your manuscript is now with our production department and you will be notified of the publication date in due course.

With kind regards,

Matt Lyles
